



**Skin temperature from the Thermal Infrared Sounder IASI**
Sarah Safieddine[1], Ana Claudia Parracho[1], Maya George[1], Filipe Aires[2], Victor
Pellet[2], Lieven Clarisse[3], Simon Whitburn[3], Olivier Lezeaux[4], Jean-Noël Thépaut[5],
Hans Hersbach[5], Gabor Radnoti[5], Frank Goettsche[6], Maria Martin[6], Marie Doutriaux-
Boucher[7], Dorothée Coppens[7], Thomas August[7], and Cathy Clerbaux[1,3]
*[1]LATMOS/IPSL, Sorbonne Université, UVSQ, CNRS, Paris, France*
*[2] LERMA, Observatoire de Paris, Paris, France*
*[3]Université libre de Bruxelles (ULB), Atmospheric Spectroscopy, Service de Chimie*
*Quantique et Photophysique, Brussels, Belgium*
*[4] Spascia, Toulouse*
*[5] ECMWF, Shinfield Park, Reading, Berkshire, RG2 9AX, UK*
*[6] Karlsruhe Institute of Technology (KIT), Eggenstein-Leopoldshafen, Germany*
*[7] European Organisation for the Exploitation of Meteorological Satellites, Darmstadt,*
*Germany*
**Abstract**
Skin temperature ($T_{skin}$) derived from infrared sensors on board satellites provides a
continuous view of Earth's surface day and night and allows for the monitoring of
global temperature changes relevant for climate trends. $T_{skin}$ from the Infrared
Atmospheric Sounding Interferometer (IASI) has not been properly exploited to date
to assess its long-term spatio-temporal variability and no current homogenous $T_{skin}$
record from IASI exists. In this study, we present a fast retrieval method of $T_{skin}$
based on an artificial neural network from a set of IASI channels selected using the
information theory/entropy reduction technique. We compare and validate our IASI
$T_{skin}$ product with that from EUMETSAT Level 2, ECMWF Reanalysis ERA5, SEVIRI
land-surface temperature products, as well as ground measurements. Our results
show good correlation between the IASI neural network product and the datasets
used for validation, with a standard deviation between 1 and 4 °C. This method can
be applied to other infrared measurements, and allows for the construction of a
robust $T_{skin}$ dataset, making it suitable for trend analysis.
**1. Introduction**

Land surface temperature, radiometric temperature, or as used hereafter, skin
temperature $T_{skin}$ depends on the energy fluxes between the surface and the
atmosphere. It is an important factor for studying the Earth's energy balance,
convection at the surface, monitoring droughts and in numerical weather prediction
(Goldberg et al., 2003; Zhou et al., 2003; Rhee et al., 2010). Although in situ
observations play a major role in measuring relevant climate change indicators, local
measurements are sparse and unevenly distributed. Global view observations are
now routinely available from remote sensors on satellites, providing data from which



climate variables, such as $T_{skin}$ can be derived using appropriate retrieval methods. The World Meteorological Organization (WMO) Global Climate Observing System (GCOS) program, aims at identifying requirements for the global climate monitoring system. It recommends 54 key variables (https://gcos.wmo.int/en/essential-climate-variables/), called Essential Climate Variables (ECVs), as the atmospheric, land, and ocean components of this monitoring system (GCOS, 2017). Near-surface temperature and skin temperature are both ECVs. In the thermal infrared spectral range, satellites do not measure the well-known thermodynamic near-surface air temperatures ($T_{2m}$); instead, they measure the skin temperature. It is called "skin" temperature since it corresponds to the radiation emitted from depths less or equal to the penetration depth at a given wavelength (Becker and Li, 1995), which can be as small as 10-20 micrometers at the ocean surface (McKeown et al., 1995). The relationship between $T_{skin}$ and $T_{2m}$ is complex: differences between $T_{skin}$ and $T_{2m}$ can reach several to ten or more degrees under cloud-free, low wind speed conditions, and is usually smaller under cloudy conditions or when solar insolation is low (Prigent et al., 2002; 2003; Good, 2016).

Satellite retrievals of skin temperatures are available from a variety of polar-orbiting and geostationary platforms carrying microwave and infrared sensors, such as the Spinning Enhanced Visible and Infrared Imager (SEVIRI) onboard the geostationary Meteosat Second Generation (Trigo et al., 2008), the Advanced Very High Resolution Radiometer (AVHRR) sensors onboard the different NOAA polar orbiting platforms and more recently on the suite of Metop satellites (Jin, 2004), the Moderate Resolution Imaging Spectroradiometer (MODIS) on board of the Terra and Aqua satellites (Wan and Li, 1997), the Atmospheric InfraRed Sounder (AIRS, Ruzmaikin et al., 2017), on board the Aqua satellite, and from the Infrared Atmospheric Sounding Interferometer (IASI) on board the three Metop satellites since 2007, 2012 and 2018 (Siméoni et al., 1997; Blumstein et al., 2004; Hilton et al., 2012).

With a polar orbit, IASI on Metop revisits all points on the Earth's surface twice a day at around 9:30 am and 9:30 pm local time. IASI is designed for numerical weather prediction, climate research and atmospheric composition monitoring (Collard et al., 2009; Clerbaux et al., 2009; Hilton et al., 2012). It measures radiances in the thermal infrared spectral range between 645 and 2760 cm$^{-1}$ corresponding to 8461 spectral channels, every 0.25 cm$^{-1}$, with an instrument response function of 0.5 cm$^{-1}$ half-width after apodization. With more than eleven years of data that are now readily available, the instrument provides more than 1.2 million radiance spectra per day with a footprint on the ground of 12 km diameter pixel (at nadir). IASI scenes are reduced by around one third when clear sky filtering (<10% cloud coverage) is applied, a necessity for accessing information at the surface. IASI has been used for atmospheric composition sounding, allowing near-real-time mapping of chemical species and aerosols, contributing to air traffic safety, and improving the understanding of atmospheric transport processes (e.g., Coheur et al., 2009; Clarisse et al., 2011; Clerbaux et al., 2015).





The interest in exploiting highly spectrally resolved IASI data to study climate
variability has been previously highlighted (Clerbaux et al., 2003; Brindley et al.,
2015; Smith et al., 2015). However, relatively little has been done to generate
systematic records for climate variables with IASI, although the spectral signature of
climate variability and $T_{skin}$ anomalies have been studied for similar instruments (e.g.
AIRS, Brindley et al., 2016; Susskind et al., 2019). The instrument is relatively new
(radiances are provided since July 2007) and the climate community is still not fully
aware of its potential. It is also computationally demanding to systematically process
the large amount of data generated by the instrument. However, since IASI is
planned for flying at least 18 years, with the 3 instruments built at the same time and
flying in constellation, continuity and stability are insured, and the potential of
constructing a long-term climate data record is becoming evident. In addition, it is
worth noting that the long-term continuation of the program is also guaranteed, as the
new generation of Infrared Atmospheric Sounding Interferometers (IASI-NG)
(Clerbaux and Crevoisier, 2013; Crevoisier et al., 2014), will be launched on three
successive Metop - Second Generation satellites within the 2022-2040 timeframe.
IASI data are disseminated by EUMETSAT (EUropean organization for the
exploitation of METeorological SATellites) (Klaes et al., 2007). It processes a $T_{skin}$
product from the series of the Metop satellites for day-to-day meteorological
applications. This $T_{skin}$ product is derived from IASI upwelling radiances but also
relies on other microwave instruments on board of Metop, particularly for cloudy
scenes. This dataset is not homogeneous in time, neither for the Level 1C (L1C),
radiances, nor for Level 2 (L2) operational products (e.g. temperature, humidity,
cloud cover, etc.). Changes occurred with evolving versions of the processing
algorithm (EUMETSAT, 2017a; EUMETSAT, 2017b), with the algorithm mostly stable
after 2016. The Metop-A L1C record has been reprocessed back in time at
EUMETSAT for the period 2007-2017, and is used in this work, and will be publically
available in summer 2019. L1C data after 2017 are not reprocessed because they
are assumed to be up to date. The Level 2 series has not yet been reprocessed back
in time, which complicates the construction of a homogeneous $T_{skin}$ data record from
IASI.
More generally, high volumes of data resulting from IASI present many challenges in
data transmission, storage, and assimilation. One of the simplest methods for
reducing the data volume is channel selection. The goal of this study is to present a
fast and reliable method developed to retrieve $T_{skin}$ from radiances using a limited set
of radiances from the newly reprocessed IASI L1C data record in the thermal infrared
in order to have a consistent and homogeneous product covering the whole IASI
sounding period.
The challenge is therefore to find the optimal set of channels from which skin
temperature can be retrieved. In the following section 2, we present an approach
based on entropy reduction (Rodgers, 1996; Collard, 2007) from which we deduce a
set of 100 channels most sensitive to skin temperature from the IASI 8461 channels.
The dataset is then used to retrieve skin temperature from IASI's cloud-free



radiances using an artificial neural network (ANN). In section 3 we validate the
product and we conclude this paper with a discussion in section 4 of the current
challenges in validation and comparison of different $T_{skin}$ products.
**2.  Data and methods**
**2.1.  Choice of IASI spectral window for $T_{skin}$ retrieval**
IASI uses three detectors to fully cover the spectral range that extends from 645 to
2760 cm$^{-1}$ (15.5 to 3.62 μm) with no gaps. To understand the spectral window that
must be used for $T_{skin}$ retrieval, we show in Figure 1, upper panel, a IASI typical
cloud-free spectra, with the corresponding Jacobian (the sensitivity of the IASI
brightness temperature to the skin temperature), as well as Signal to Noise Ratio
(SNR), and radiometric noise. The recorded spectrum, with an example shown in red
in the upper panel of Figure 1, in brightness temperature units, exhibits signatures
associated with spectroscopic absorption/emission lines of molecules present along
the optical path between the Earth's surface and the satellite detectors. From these
spectra, geophysical data such as temperature profiles and atmospheric
concentrations of trace gases can be derived from selected spectral windows.
Channels that are candidates for $T_{skin}$ retrieval are therefore located in spectral
windows with little interference from other absorbing/emitting molecules, and are also
those where the $T_{skin}$ Jacobians (blue line in upper panel) are the highest. These are
the spectral ranges before and after the ozone band, i.e., 800-1040 cm$^{-1}$ and 1080-
1150 cm$^{-1}$, the small spectral window after the water vapor continuum at ~2150 cm$^{-1}$
and the spectral range > 2400 cm$^{-1}$.

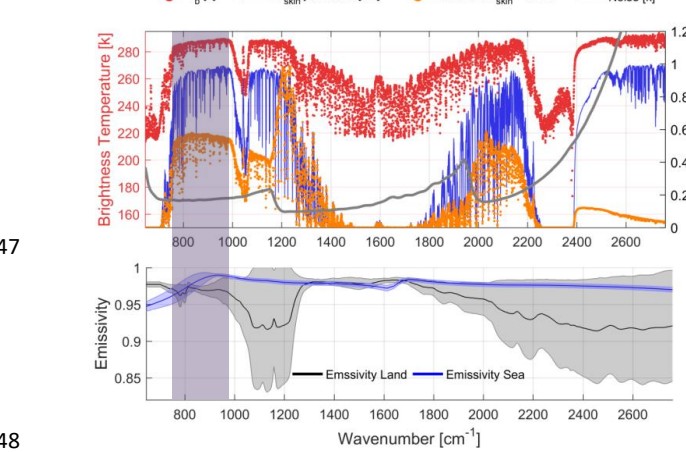

Figure 1. *Upper panel:* brightness temperatures for a random cloud-free spectrum
(red). On the right axis, $T_{skin}$ Jacobians in k/k (dark blue), signal-to-noise ratio
obtained for a variation of $T_{skin}$ of 0.1 k (orange), and IASI radiometric noise spectrum
(grey), calculated using RTTOV (Saunders et al., 2018). *Lower panel:* Average
emissivity over land (black), and sea (blue), with the corresponding standard



deviation in shaded colors around the lines. The shaded vertical strip shows the
spectral window used for $T_{skin}$ retrievals in this study.

The window > 2400 cm$^{-1}$, as well as that around ~2150 cm$^{-1}$ may be contaminated by
solar radiation during the day. In terms of SNR, the very important values of the
radiometric noise at >2400 cm$^{-1}$ induces a low value of the SNR. The spectral band
at ~2150 cm$^{-1}$ presents a slightly weaker performance than the spectral ranges
around the ozone absorption band. These two spectral bands (~2150 and > 2400 cm$^{-1}$
$^{1}$) are therefore not critical for the $T_{skin}$ retrieval and are discarded.

The lower panel of Figure 1 shows the average emissivity over land (in black) and
sea (in blue). Emissivity is needed to calculate $T_{skin}$ from the radiative transfer
equation. In this work, we want to use a method without prior assumption on
emissivity. Nevertheless, we should be careful with our choice of channels' emissivity
in our selected spectral window. We can see that on the right of the ozone band,
around 1100-1200 cm$^{-1}$, the variability of the emissivity, especially over land is much
more important than the window between 750 and 970 cm$^{-1}$, shown in the shaded
rectangle in Figure 1, where also the noise is smaller, and the SNR higher. This
makes this spectral window the best candidate for $T_{skin}$ retrieval.

**2.2.    Channel Selection based on Entropy Reduction**

We use an iterative method where channels are selected based on their ability to
reduce the uncertainty of retrieving temperature. It was proposed by Rodgers (1996,
2000), evaluated for IASI by Rabier et al. (2002) and applied by Collard et al. (2007)
to Numerical Weather Prediction (NWP).
The method has been rigorously studied and relies on evaluating the impact of the
addition of single channels on a theoretical retrieval based on a figure of merit, such
as the Entropy Reduction (ER), used in this study, and defined as follows:

$$ER = \tfrac{1}{2} log_2 \left( \tfrac{B}{A} \right).$$    Eq. (1)

$ER$ measures the probabilities of the ensemble of possible states in the retrieval, and
is maximal if all the states have an equal probability. The lower the entropy of the
ensemble, the better the retrieval. The channel that reduces this entropy emphasizes
a particular state of the retrieval. Entropy reduction is a metric derived from
information theory. In Eq. (1), $A$ is the analysis-error covariance matrix, and $B$ is the
background/*a priori* error covariance matrix, with:

$$A = (B^{-1} + H^T R^{-1} H)^{-1},$$    Eq. (2)

Where $H$ is the Jacobian matrix of $T_{skin}$ and $R$ the covariance matrix of instrumental
and radiative transfer noises. "External variables" such as water vapor or ozone can





contaminate a given candidate $T_{skin}$ channel by absorbing in the targeted spectral
range. This might affect the selection, and introduces an error that should be added
to the $A$ matrix (Aires et al. 2016, Pellet and Aires, 2016). If those errors were not
included in the background $B$ matrix, the quality of the selected channels might be
artificially over-estimated. When this contaminating effect is defined explicitly, Eq. (2)
is updated to:

$$A_{V^{-1}} = B_V^{-1} + H_V^t \cdot (R + H_v \cdot B_v \cdot H_v^t)^{-1} \cdot H_V \qquad \text{Eq. (3)}$$

Where $V$ is the variable to be retrieved ($T_{skin}$) and $v$ is the external variable (e.g.
ozone or water vapor). This equation is valid by making some assumptions, in
particular that no correlation between $V$ and $v$ exists and that the impact of this
external variable contamination on the channel is an error with Gaussian distribution
with covariance matrix $H_v^t \cdot B_v \cdot H_v$ .
In most channel selection analyses, the errors from external variables (such as that
of relative humidity or ozone) are not taken into account in the measurement of the
information content of the candidate channel. Collard (2007) attempted to take into
account the effects of trace gases not included in the radiative transfer simulation by
inflating the observation errors for channels that showed sensitivity to the missing
species. A more complete approach was adopted by Ventress and Dudhia (2014),
who used climatological variability of atmospheric constituent species to model their
effect on the radiances during the channel selection process.
In this work, we explicitly consider the contamination effect in the selection process of
dedicated $T_{skin}$ related-channels. This refined methodology improves the
representation of contamination effects from atmospheric species and therefore the
reliability of the background error covariance matrix $B$. This matrix $B$ characterizes
the quality of the a priori information and varies in space and time in order to account
for its complex state-dependence. For this work, we derive a Gaussian $B$ matrix as:
$B = Cov(x, y) = Corr(x, y) \cdot \sigma(x) \cdot \sigma(y)$, where $\sigma$ is the standard deviation of each of
the variables to consider ($T_{skin}$, atmospheric temperature, relative humidity, and
ozone) at the vertical level x and y. An uncertainty of $\sigma$ = 2 k is chosen for $T_{skin}$ as
done in the study by Collard (2007). The covariance and correlation matrices of the
background errors for relative humidity and ozone are calculated based on the widely
used assumption that humidity (or ozone) error correlation between the vertical layers
is close to the actual associated humidity (or ozone) correlation. We choose to have
the covariance matrices $B$ for humidity and ozone based on the raw humidity and
ozone correlation matrices, and an error variance ($\sigma^2$) of 20% for humidity, and 30%
for ozone on each vertical atmospheric layer. As humidity and ozone can impact $T_{skin}$
channel selection, error along the vertical is needed for $T_{skin}$ retrieval.
An iterative method (Rodgers, 1996) is used to forwardly select the most informative
channels. In order to speed up the computations, an efficient algorithm was





developed assuming that the observation errors are uncorrelated between channels.
However, as the IASI radiances are apodized, and thus have highly-correlated errors
between adjacent channels, a channel is not selected if its immediate neighbor is
already chosen (Collard, 2007).
The iterative procedure is initialized with $A_0 = B$, and the Jacobian $H$ (which is
constant during the iteration) is normalized with the instrumental noise covariance
matrix $R$, as follows: $H' = R^{-1/2}H$.
According to Rodgers (1996), the updated analysis error covariance matrix at each
iteration step $i$ can be calculated from the previous step $i - 1$ as follows:
$$A_i = A_{i-1} - \frac{(A_{i-1}h')(A_{i-1}h')^T}{1 + (A_{i-1}h')^T h'}$$


Where $h'$ is the column vector equal to the row of $H'$ for the candidate channel.
The ER change between two iterations can now be written as:
$$\delta ER = \frac{1}{2} log_2(1 + h'^T A_{i-1} h')$$

At each step, the channel that has the largest information content (measured as a
reduction of the entropy of the corresponding $T_{skin}$ retrieval when the candidate
channel is used) is selected, given the information content of the previously selected
channel(s). The channel selection starts with no channel selected, and sequentially
chooses the channel with the highest information content in complement to the
information from all the previously selected channels.
The spectra and Jacobians used in this study were simulated using the last version of
the Optimum Spectral Sampling (OSS) radiative transfer model (Moncet et al., 2008),
using the Thermodynamic Initial Guess Retrieval (TIGR3) database (Chevallier et al.,
1998), and more detailed description on the atmospheric profiles, the radiative
transfer code, and the Jacobians, can be found in Pellet and Aires (2018).
Here, a channel selection is only performed over the spectral window of $T_{skin}$ retrieval
as was discussed in section 2.1, and is shown in Figure 2. The IASI spectral window
was divided into 100 spectral subsets and a channel selection was applied to each.
Using this method, we selected the best 100 channels in terms of information content
and the resulting selection is listed in Table 1 and presented in Figure 2. The figure
shows that most of the selected channels are between 760 and 980 cm$^{-1}$. However,
few channels are also selected for wavenumbers < 760 cm$^{-1}$ since in this part of the
spectrum, the atmospheric vertical levels are very correlated to one another and
therefore information on the surface exists in these channels.

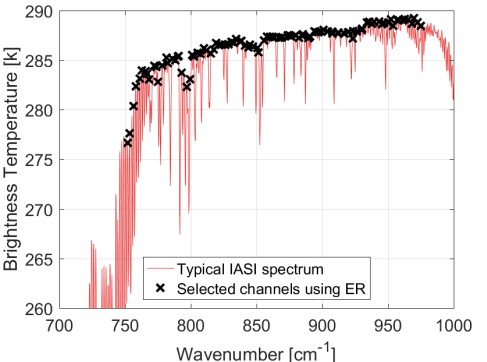

Figure 2. The location of the 100 selected channels using the ER method displayed on a IASI spectrum.









Table 1. The 100 channels used for $T_{skin}$ retrieval selected using the Entropy
Reduction (ER) method. Channels are sorted from the highest to the lowest
information content (top to bottom and left to right).

| Channel | Wavenumber (cm⁻¹) | Channel | Wavenumber (cm⁻¹) | Channel | Wavenumber (cm⁻¹) | Channel | Wavenumber (cm⁻¹) |
|---|---|---|---|---|---|---|---|
| 1300 | 969.75 | 1038 | 904.25 | 853 | 858.00 | 682 | 815.25 |
| 1282 | 965.25 | 1100 | 919.75 | 984 | 890.75 | 582 | 790.25 |
| 1249 | 957.00 | 1001 | 895.00 | 862 | 860.25 | 630 | 802.25 |
| 1272 | 962.75 | 1321 | 975.00 | 771 | 837.50 | 625 | 801.00 |
| 1254 | 958.25 | 1209 | 947.00 | 759 | 834.50 | 574 | 788.25 |
| 1294 | 968.25 | 1069 | 912.00 | 752 | 832.75 | 584 | 790.75 |
| 1230 | 952.25 | 997 | 894.00 | 797 | 844.00 | 547 | 781.50 |
| 1164 | 935.75 | 1070 | 912.25 | 745 | 831.00 | 551 | 782.50 |
| 1267 | 961.50 | 921 | 875.00 | 775 | 838.50 | 565 | 786.00 |
| 1194 | 943.25 | 962 | 885.25 | 801 | 845.00 | 516 | 773.75 |
| 1179 | 939.50 | 1051 | 907.50 | 714 | 823.25 | 510 | 772.25 |
| 1222 | 950.25 | 940 | 879.75 | 706 | 821.25 | 593 | 793.00 |
| 1311 | 972.50 | 916 | 873.75 | 698 | 819.25 | 534 | 778.25 |
| 1086 | 916.25 | 1114 | 923.25 | 844 | 855.75 | 484 | 765.75 |
| 1157 | 934.00 | 950 | 882.25 | 726 | 826.25 | 472 | 762.75 |
| 1172 | 937.75 | 869 | 862.00 | 810 | 847.25 | 488 | 766.75 |
| 1142 | 930.25 | 1237 | 954.00 | 736 | 828.75 | 494 | 768.25 |
| 1203 | 945.50 | 926 | 876.25 | 824 | 850.75 | 466 | 761.25 |
| 1018 | 899.25 | 961 | 885.00 | 691 | 817.50 | 619 | 799.50 |
| 1141 | 930.00 | 875 | 863.50 | 669 | 812.00 | 609 | 797.00 |
| 1009 | 897.00 | 979 | 889.50 | 661 | 810.00 | 521 | 775.00 |
| 1089 | 917.00 | 889 | 867.00 | 786 | 841.25 | 454 | 758.25 |
| 1115 | 923.50 | 899 | 869.50 | 827 | 851.50 | 447 | 756.50 |
| 1025 | 901.00 | 897 | 869.00 | 642 | 805.25 | 435 | 753.50 |
| 1126 | 926.25 | 1052 | 907.75 | 650 | 807.25 | 429 | 752.00 |


**2.3. Artificial Neural Network for $T_{skin}$ retrievals**
Artificial neural networks (ANN) method is used to approximate the complex radiative
transfer function that maps the radiances to skin temperature. The training dataset is
constructed out of clear-sky (cloud cover <10%) Level 1C (L1C) IASI radiances over
the 100 channels selected in section 2.2. We train our ANN with these IASI radiances
but test two different datasets as output/target. In the first, we use the $T_{skin}$ from the
ERA5 reanalysis (Copernicus Climate Change Service, 2017) as output/target. $T_{skin}$
is very sensitive to surface properties, which depend on local meteorological
conditions (Good, 2016). To this end, a few dedicated ERA5 experiments were
performed at ECMWF at a 12-minute time-step (as opposed to the publicly released





hourly $T_{skin}$ product), each spanning a couple of days. The aim of these experiments
is to increase the temporal resolution and therefore increase the performance of the
neural network obtained. Four days in January and June 2018 are used for the
training to represent seasonality. We interpolate ERA5 space/time grid to IASI's
observations (at 9:30 AM and PM local time). We provide more information on the
ERA5 reanalysis in section 3. The resulting training dataset is formed out of around
$5.9 \times 10^5$ scenes.
In the second training, we use EUMETSAT L2 $T_{skin}$ product as target. EUMETSAT
$T_{skin}$ is derived from Metop observations and the IASI instrument. They are therefore
collocated in space and time. Since major and minor updates on the processing
algorithms of the L1C and L2 EUMETSAT product took place in the past 10 years
(EUMETSAT, 2017a; 2017b), the ANN training in this study uses a recent and
coherent year, 2018. To represent the seasonal variability, scenes from January 1$^{st}$,
April 1$^{st}$, July 1$^{st}$, and October 1$^{st}$ 2018 are used. The resulting training dataset is
formed out of around $9 \times 10^5$ scenes for EUMETSAT. More information on the
EUMETSAT $T_{skin}$ product is provided in section 3.
Since IASI has more frequent overpasses at the poles (given its polar orbit), a
weighting function is applied to equally distribute the number of scenes around the
globe. The training is done using mini-batches with a maximum of 10.000 epochs to
train. The ANN has 2 hidden layers with 4 nodes, and a network training function that
updates weight and bias values according to Levenberg-Marquardt optimization.
The neural network learns how to associate any set of radiances to a corresponding
skin temperature. The feasibility of using ANN to $T_{skin}$ retrieval has been shown for
instance by Aires et al. (2002) for IASI, and has also been performed to tackle
various problems in atmospheric remote sensing (Blackwell and Chen, 2009; Hadji-
Lazaro et al., 1999; Whitburn et al., 2016; Van Damme et al., 2017). In the following
"$T_{ANN}$" refers to the product developed in this study using artificial neural networks
from IASI radiances.
Figure 3 shows the training results when the $T_{ANN}$ is compared with the $T_{ERA5}$ dataset
is used for the training, and in Figure 4 when the $T_{EUMETSAT}$ is used for the training.
We achieve a good agreement with a standard deviation of 2.2 and 1.6 respectively
and a correlation coefficient close to 1. The largest differences are for points located
near the poles and at high altitudes. One of the reasons behind the discrepancies in
mountainous regions is the general under-representation of the orography in global
numerical weather prediction (NWP) and climate models, due to their limited
horizontal resolution. Orographic features exert drag and its correct representation in
models is extremely challenging. The incorrect representation of drag might lead to
errors in simulating surface properties and might be responsible for the bias seen in
mountainous regions (ECMWF, 2016).   Moreover, with altitudes and variable
emissivity in these regions, the neural network fails (to some extent) to properly map
the altered radiances due to surface inhomogeneity into a correct skin temperature.



Figures 3b and 4b also show how the difference between the two products is lowest
over the sea, which can suggest the robustness of this method, in particular for sea
skin temperature analysis.

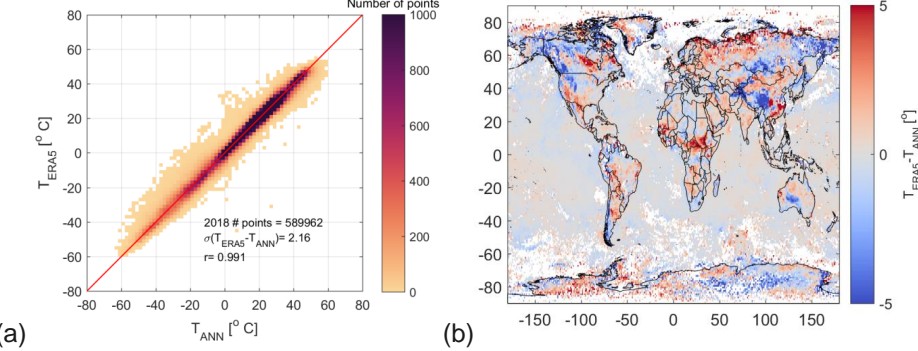

(a)                                                        (b)

Figure 3. Neural network performance when trained with ERA5 data: (a) scatterplot
and correlation, (b) gridded and averaged spatial comparison.


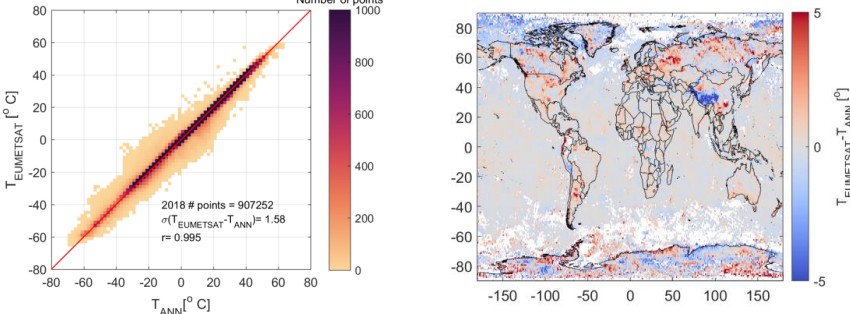


Figure 4. Neural network performance when trained with EUMETSAT data: (a)
scatterplot and correlation, (b) gridded and averaged spatial comparison.


**2.4.    Datasets used for validation**

We compare the $T_{ANN}$ from the two training datasets to the EUMETSAT L2 product,
the ECMWF ERA5 reanalysis, the SEVIRI satellite retrieval, and ground
observations. We described each briefly hereafter.
***2.4.1. EUMETSAT $T_{skin}$ product***
Meteorological L2 data from EUMETSAT (August et al., 2012) are provided for nearly
all IASI observations by deriving $T_{skin}$ primarily from IASI for cloud-free scenes and
using the Advanced Microwave Sounding Unit (AMSU), and the Microwave Humidity





Sounder (MHS) for cloudy scenes (EUMETSAT, 2017a; 2017b). AMSU and MHS are multi-channel microwave radiometers, which measure radiances in 15 and 5 discreet frequency channels respectively, and provide information on various aspects of the Earth's atmosphere and surface. They both can be used for cloud-contaminated scenes, since they are synchronized with IASI's scanning. The algorithm is based on optimal estimation. Since the algorithm uses on instruments on board of Metop, the IASI ANN cloud-free radiances used in this study are also co-localized in space and time.

### 2.4.2. ERA5 $T_{skin}$ product

In the framework of the ECMWF latest reanalysis (ERA5) (Hersbach and Dee, 2016; Hersbach et al., 2018; Copernicus Climate Change Service, 2017), skin temperature is defined as the temperature of the surface at radiative equilibrium. It is derived from the surface energy balance within the land model in ERA5 and no assimilation of surface skin temperature observations takes place. Radiances on the other hand, are assimilated. The surface energy balance is satisfied independently for each tile by calculating its skin temperature. The skin layer represents the vegetation layer, the top layer of the bare soil, or the top layer of the snow pack. In order to calculate the skin temperature, the surface energy-balance equation is linearized for each tile leading to an expression for the skin temperature (ECMWF, 2016). Over the ocean, the sea surface temperature (SST) is specified from an analysis provided by the Operational Sea Surface Temperature and Ice Analysis (OSTIA, McLaren et al., 2016) from September 2007 and prior to that date from the Met Office Hadley Centre HadISST2 product (Hirahara et al., 2016). The SST analysis is a blend of satellite retrievals and in situ observations from ships, and ensures a detailed horizontal distribution from satellite data anchored to the sparse ship observations. The resulting SST fields are therefore calibrated as if they are ship observations and therefore they represent bulk SST fields (i.e. measured a few meters deep). Since the ocean skin temperature (<1 mm thickness) might be cooler than the SST because of the turbulent and long wave radiative heat loss to the atmosphere, parameterizations of different near surface ocean effects are included in the code (ECMWF, 2016).

### 2.4.3. SEVIRI $T_{skin}$ product

The Spinning Enhanced Visible and Infrared Imager (SEVIRI) onboard the geostationary Meteosat Second Generation (MSG) satellite scans the Earth surface every 15 min and provides observations in 12 spectral channels with a sampling distance of 3 km at nadir. MSG's nominal position at 0° longitude and SEVIRI's large field of view (up to 80° zenith angle) allows for frequent observations of a wide area encompassing Africa, most of Europe and part of South America (Schmetz et al., 2002).

The land surface temperature (LST) product (LSA-001) used for validation here
(Trigo et al., 2011; Freitas et al., 2010) is retrieved by the EUMETSAT Land Surface
Analysis Satellite Application Facility (LSA SAF) with the generalized split-window
method, which requires land surface emissivity as input data. IASI and SEVIRI data
are spatially co-located when observations from each instrument are less than 5
minutes apart, and within 0.25 degrees in longitude and latitude.

### 2.4.4 Ground observations

The ground observations are from Gobabeb wind tower, Namibia (23.551° S 15.051°
E, location shown in Figure 7, Göttsche et al., 2016). Gobabeb station is located on
the large and homogenous Namib gravel plains (Göttsche and Hulley, 2012).
Göttsche et al. (2013) showed that the station $T_{skin}$ is representative for an area of
several 100 km² , making it suitable for validation with satellite measurements. $T_{skin}$ is
obtained once per minute with the station's core instrument, an infrared precision
radiometer (Heitronics KT15.85 IIP) measuring radiances between 9.6 and 11.5 µm.
The temperature resolution is given as 0.03 K with an uncertainty of ±0.3 K over the
relevant range, and high stability with a drift of less than 0.01% per month (Goettsche
et al., 2013).

## 3. Results

To validate the $T_{ANN}$ product, the month of June 2016 is chosen. Since we train our
neural network with 2018 data, 2016 is a good choice and data is readily available for
this year. $T_{ANN}$ is calculated from the two ANNs obtained in section 2 by applying it to
each set of 100 radiances retrieved from IASI for all cloud-free observations in June
429  2016.

### 3.1.  Validation of the $T_{ANN}$ obtained from the ERA5 neural network

Figure 5 shows the comparison of the $T_{ANN}$ IASI obtained from the training of IASI
radiances with ERA5 12-minute data. We start by performing the validation with the
EUMETSAT, ERA5, and SEVIRI $T_{skin}$ datasets. The upper panel shows the
correlation plots, superimposed with the average difference by latitude in red. $T_{ANN}$
from IASI compares best with the EUMETSAT $T_{skin}$ product (standard deviation
σ=1.83˚C), which is plausible since it is also obtained from IASI radiances.
Comparison with ERA5 also shows a correlation close to 1, and σ=2.17˚C. The
largest differences for both EUMETSAT and ERA5 products are found around the
poles, which are probably due to the sensitivity of radiances to surface properties and
to orography-related physical processes in the ECMWF model as previously
discussed. Moreover, ERA5 data are at 0.25˚x0.25˚ resolution (native horizontal
resolution of ERA5 is ~31km) and are interpolated to the center of the IASI pixel
observation, which might correspond to a different surface type and might lead to
differences in temperatures. For the comparisons between $T_{ANN}$ IASI and $T_{skin}$



SEVIRI a standard deviation of σ=3.78 K is determined with the largest differences
over the Arabian Peninsula. For large viewing angles, in particular near the edge of
the Meteosat disk (such as the Arabian Peninsula), the uncertainty of SEVIRI $T_{skin}$ is
high (Freitas et al., 2010). A study by Trigo et al. (2015) reported similar to larger cool
biases in the rest of the domain between the ECMWF model data and SEVIRI,
especially over semiarid regions, such as North Africa, Sahara, and Namibia. In the
rest of the domain, the two datasets agree reasonably well.

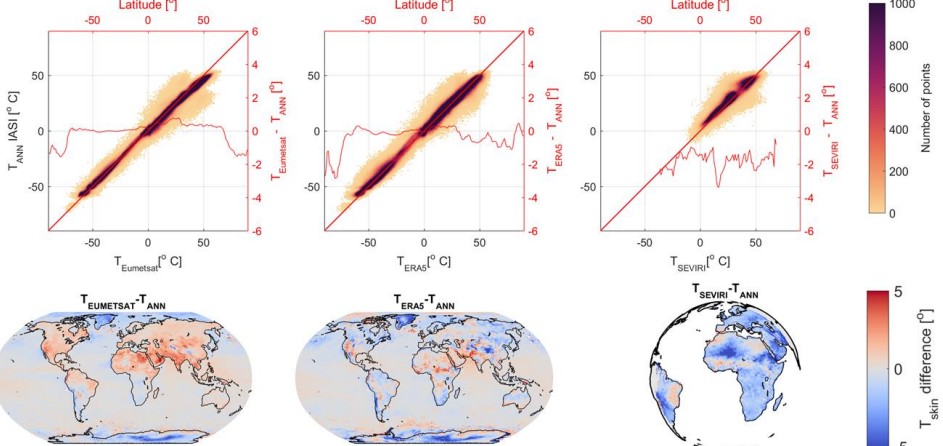

Figure 5. Validation of the $T_{skin}$ ANN product ($T_{ANN}$) from the neural net training of
IASI radiances with ERA5, with products from EUMETSAT, ERA5 and SEVIRI, for
June 2016. Upper panel: correlation plots weighted with the number of co-localized
observations during one month. Lower panel: gridded and averaged spatial
difference [T − $T_{ANN}$]. For day + night data: σ ($T_{EUMETSAT}$ − $T_{ANN}$) =1.83, σ ($T_{ERA5}$ −
$T_{ANN}$) =2.17, σ ($T_{SEVIRI}$ − $T_{ANN}$) =3.78. The total number of points for the global
comparison is 8.2 x $10^6$ and 4.96 x $10^5$ for the SEVIRI comparison.


While this paper focuses on validating IASI $T_{ANN}$, inter-comparisons between the
different products (ERA5 with EUMETSAT L2 or EUMETSAT L2 with SEVIRI, etc.)
are valuable for assessing their differences. Figure 6 shows the box plot of these
inter-comparisons, with the absolute bias and standard deviation of the comparison
between the products. We perform inter-comparisons for day- and night-times
separately. At nighttime, the absence of solar illumination allows a direct comparison
of the skin temperature retrieved or modelled from different instruments. It can be
seen that the $T_{ANN}$ product developed in the framework of this study is within the
range of biases among the other products comparison.



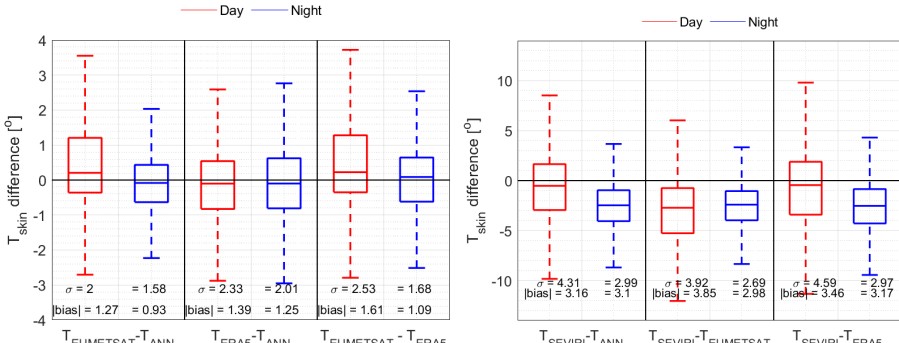


Figure 6: Boxplot of the June 2016 inter-comparison of the different $T_{skin}$ products
used in this study. Since the matching with SEVIRI leads to fewer co-localized data
points covering the SEVIRI disk, they are shown on a separate figure on the right.
The central mark indicates the median, and the bottom and top edges of the box
indicate the 25th and 75th percentiles.

Figure 6 shows that the night observations of $T_{ANN}$, $T_{EUMETSAT}$ and $T_{ERA5}$ seem to
agree better with each other, an expected result and also detected for other satellite
data (August et al., 2012; Martin et al., 2019).
Comparison with SEVIRI shows a consistent negative bias during the night when
compared to $T_{ANN}$, $T_{EUMETSAT}$ and $T_{ERA5}$. Several studies (e.g., Garand, 2003; Zheng
et al., 2012) already reported cold biases between SEVIRI and other $T_{skin}$ products.
For the ECMWF model, the cold bias over land was identified for a previous version
of the model by Trigo and Viterbo (2003) and for a more recent version by Trigo et al.
(2015). A misrepresentation of surface energy fluxes, either because of deficiencies
in the parameterization of aerodynamic resistances, or in the partitioning between
latent and sensible heat fluxes are frequent causes of these deviations (Trigo et al.,
2015). The EUMETSAT $T_{skin}$ product seem to agree the least with SEVIRI both
during the day and the night, similar to what was reported by August et al., 2012. The
standard deviation is the largest during the day, since the comparison is affected by
the different Sun–surface–instrument geometries. Shadows due to orography or
vegetation for example change in daytime with varying SEVIRI and Metop scan angle
(August et al., 2012).

We also use station data for June 2016 for validating the $T_{ANN}$ product. This site is
chosen in order to minimize complications from spatial scale mismatch between
ground-based and satellite sensors. IASI cloud-free data was co-localized in space
and time (within 1 minute of the station data). The spatial matching is done around
0.5° of a validation site [15.17°E, 23.18°S] which location is shown in shown in panel
(a). This validation location was chosen because it is close of the station site and is
representative of the same gravel plain surface, yet, away from the sand dunes
limiting the station. The location of the station and the corresponding IASI





observations is shown in Figure 7, panel (a). The total number of coincident IASI data
points around this area is 82. The validation of the $T_{ANN}$ with in-situ $T_{skin}$ is shown in
Figure 7 panels (b), (c) and (d).


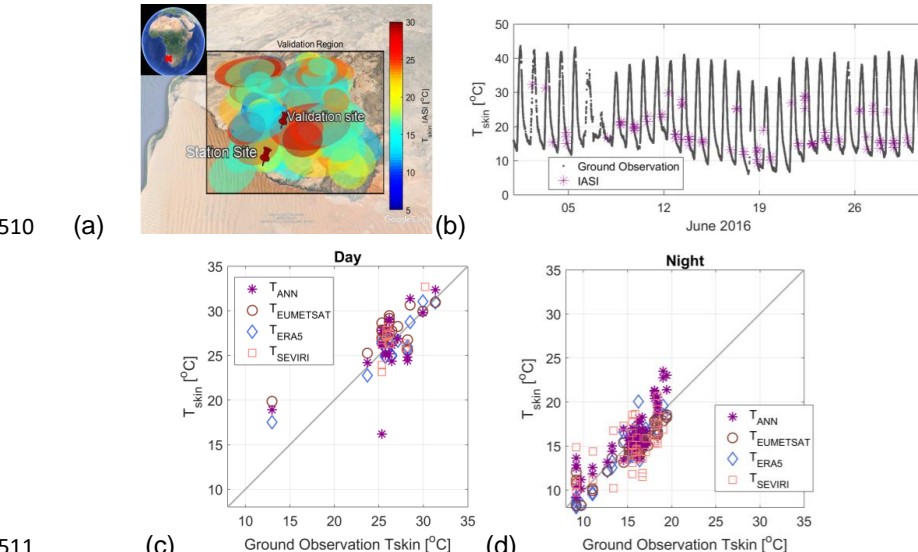

(a)                                    (b)
(c)                               (d)
Figure 7: Comparison of IASI $T_{ANN}$ with ground observations at Gobabeb: (a) station
location and the 82 coincident IASI observations in June 2016 around the validation
site chosen so all IASI observations fall in the gravel plains; (b) Diurnal variation of
$T_{skin}$; (c) $T_{ANN}$ versus in-situ $T_{skin}$ during the day; and (d) during the night.

Panel (b) of Figure 7 shows the strong diurnal variation of $T_{skin}$ observed at Gobabeb.
The IASI data are either from the morning (~9-10 am depending on the satellite
swath) or evening overpass (~9-10 pm): they are therefore always separated by ~12
hours.
Day and night correlation coefficients are > 0.9. Table 2 lists how the different
datasets used for validation compare to ground measurements. During the day, $T_{ANN}$
agrees the least with the station data, driven by the one point in Figure 7 panel (c)
that has the largest bias. At night, $T_{ANN}$ comparison with ground measurements is
better, so is the comparison with other datasets, as also seen in Figure 7. Absolute
biases mostly range between 0 and 2 K, which is similar to the $T_{skin}$ spatial variability
around Gobabeb station determined with detailed measurements carried by
Goettsche et al. (2013). Comparison with other satellite measurements shows a
general bias between -2 and 5 kelvins in summer months (Martin et al., 2019).

Table 2. Correlation coefficient, standard deviation, and absolute relative bias (%),
between ground based $T_{skin}$ and the different datasets used in this study

|  | Day | Night |
| --- | --- | --- |





|  | Standard deviation [º] | Absolute bias [º] | Standard deviation [º] | Absolute bias [º] |
|---|---|---|---|---|
| **T$_{ANN}$ – ground** | 3.12 | 2.14 | 1.67 | 1.41 |
| **T$_{EUMETSAT}$ – ground** | 1.99 | 2.03 | 1.00 | 1.06 |
| **T$_{ERA5}$ – ground** | 1.57 | 1.18 | 1.06 | 1.01 |
| **T$_{SEVIRI}$ – ground** | 1.67 | 1.50 | 2.45 | 2.09 |


### 3.2. Validation of the T$_{ANN}$ obtained from the EUMETSAT neural network

The validation presented hereafter is similar to what was shown in Figures 5, 6, and
7, and the discussion used for the discussion of the biases in those figures applies
here too. Since the T$_{ANN}$ validated here is derived from the EUMETSAT L2 product, it
compares best with it as it is seen in Figure 8.

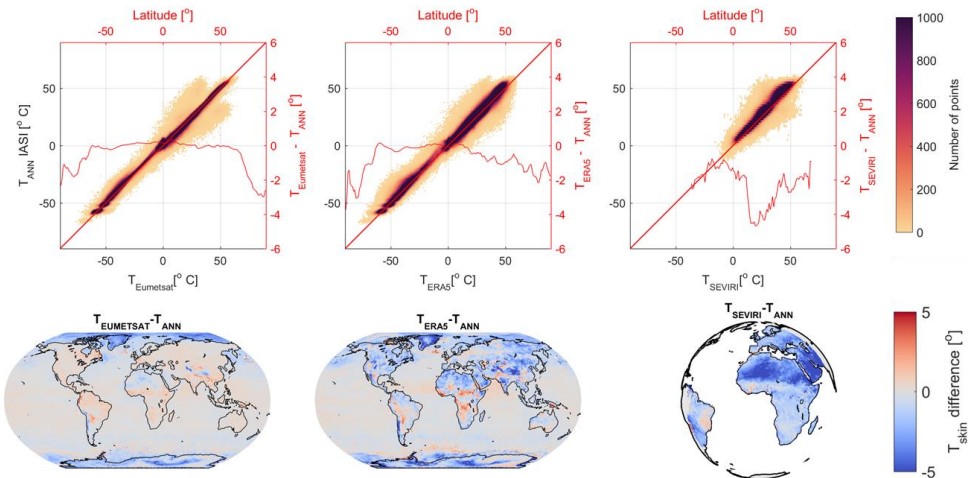


Figure 8. Same as Figure 5 but for T$_{ANN}$ derived from the EUMETSAT T$_{skin}$ neural
network. For day + night observation: σ (T$_{EUMETSAT}$ − T$_{ANN}$) =1.56, σ (T$_{ERA5}$ − T$_{ANN}$)
=2.41, σ (T$_{SEVIRI}$ − T$_{ANN}$) =3.67. The total number of points for the global comparison
is 8.2 x 10$^6$ points and 4.96 x 10$^5$ for the SEVIRI comparison.


Figure 9 is derived from data used in Figure 8, but separated into day and night, and
includes the inter-comparison of the different products with each other. The y-axis
limit is kept the same as in Figure 6 for quick comparisons. Again, T$_{ANN}$ in this case
agrees best with the EUMETSAT product, but also shows a similar good
performance when compared to other datasets.



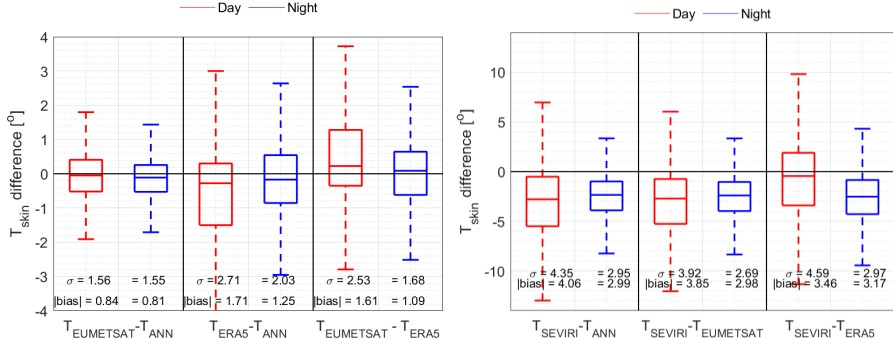

Figure 9: Boxplot of the June 2016 inter-comparison of the different $T_{skin}$ products used in this study. Since the matching with SEVIRI leads to fewer co-localized data points covering the SEVIRI disk, they are shown on a separate figure on the right. The central mark indicates the median, and the bottom and top edges of the box indicate the 25th and 75th percentiles.

Finally, comparison with ground observation in Figure 10 shows a better performance of $T_{ANN}$ than what was presented in Figure 6. Table 3 hereafter details the day and night biases where we can see that the $T_{ANN}$ in this case agrees better with ground measurements that what we presented in Table 2.

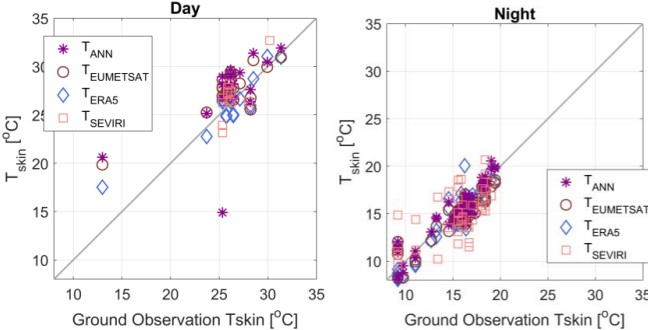

Figure 10: Comparison of IASI $T_{ANN}$ derived from EUMETSAT neural network with ground observation at Gobabeb. *Left panel:* day, *right panel:* night.



Table 3. Correlation coefficient, standard deviation, and absolute relative bias (%),
between ground based $T_{skin}$ and the different datasets used in this study

| | Day | | Night | |
|---|---|---|---|---|
| | Standard deviation [º] | Absolute bias [º] | Standard deviation [º] | Absolute bias [º] |
| $T_{ANN}$ – ground | 3.37 | 2.61 | 1.05 | 0.85 |
| $T_{EUMETSAT}$ – ground | 1.99 | 2.04 | 1.00 | 1.06 |
| $T_{ERA5}$ – ground | 1.57 | 1.18 | 1.06 | 1.01 |
| $T_{SEVIRI}$ – ground | 1.67 | 1.50 | 2.45 | 2.09 |

## 4. Discussion and Conclusions
Satellite data are able to provide systematic global temperature data, at least in
cloud-free areas, from pole to pole on a regular basis. EUMETSAT has been
updating different versions of algorithms to retrieve the skin temperature from IASI,
and at the same time, relying on different instruments (particularly for cloudy scenes)
to derive a $T_{skin}$ product. Consequently, no homogenous consistent IASI $T_{skin}$ record
exists to date. In this study, we derive a $T_{skin}$ product using Metop-A IASI L1C
radiances. The first challenge is to find the channels with access to surface
information. To this end, we present a method based on entropy reduction, to find the
channels with the highest information content in skin temperature. An efficient and
fast IASI retrieval algorithm based on artificial neural networks is then used to
calculate $T_{skin}$ from the upwelling IASI radiances. While empirical methods using ANN
can deal with hundreds to thousands of channels (Aires et al., 2002), we show in this
study how ANN and channel selection can be used to retrieve $T_{skin}$, making this
method fast and reliable for near real-time application, as well as to reprocess more
than 11 years of IASI data. In this study, we perform two ANN trainings in 2018 with
IASI radiances as input and we use two distinct datasets for two separate trainings.
In the first, a dedicated ERA5 12-minute simulation is used as output, and in the
second EUMETSAT L2 data is used as output. Each of the resulting neural networks
is then applied for a different year (2016) and validated. Our results show the
potential of ANN in mapping radiances globally and locally to skin temperature. We
show how both neural networks perform similarly well when compared to other
datasets, with the EUMETSAT-derived network performing better (in particular during
nighttime) when it is compared to ground station $T_{skin}$. To compare the two products
obtained from the two neural networks, we show in Figure 11 the daily variation of the
skin temperature in 2017, for the Northern Hemisphere in the left panel and the
Southern Hemisphere in the right panel. Generally, all datasets agree well with one
another, with $T_{ANN}$ obtained from the ERA5 $T_{skin}$ product closer to the latter (which is
expected) same as $T_{ANN}$ obtained from the EUMETSAT L2 $T_{skin}$ product is closer to
the actual EUMETSAT $T_{skin}$ product.



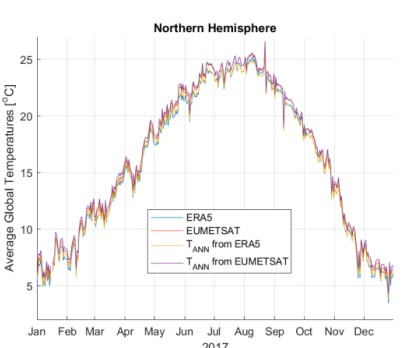
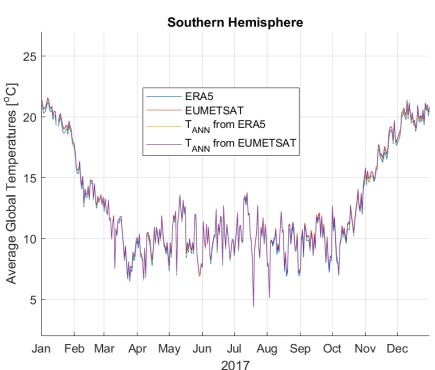


Figure 11. Daily averaged $T_{skin}$ from the different global datasets used (ERA5 and EUMETSAT L2) and produced ($T_{ANN}$ obtained from ERA5 and EUMETSAT L2) in this study.

More generally, retrieval of $T_{skin}$ from space measurements faces many challenges. First, the $T_{skin}$ calculation from the radiance within the radiative transfer equation is an ill-posed problem. The solution of the radiative transfer equation requires the simultaneous knowledge of two unknowns: $T_{skin}$ and the surface emissivity. This is generally solved with the assumption of a good initial guess to constrain the solution (Aires et al., 2001; Paul et al., 2012) and a rapid and accurate direct transfer model (Rodgers, 1976). Since the observed radiance spectra are affected by the surface properties, using it as input to the ANN takes emissivity knowledge into account.

Second, infrared retrievals are only available under clear-sky conditions, reducing the amount of global data by roughly one third. This study has been performed with data from IASI on Metop A, and it implies that with IASI on Metop B and Metop C, the global coverage can be enhanced.

Third, validation and inter-comparison between different products are challenges that not only bound to this study. The diversity in sensor characteristics and sensor-specific skin temperature retrieval algorithms, as well as the different challenges facing current NWP models, make it difficult to homogenize different skin temperature products for proper comparison. Moreover, for polar-orbiting satellite products, inter-comparison between different $T_{skin}$ satellite products is challenging since the crossing times of the satellites, and the shape of the field of view are different. For example, MODIS (with overpass time at 10:30 am/pm on TERRA) and MODIS and AIRS, on the AQUA platform (with an overpass time of 1:30am/pm), both offer a good skin temperature product. IASI on the other hand, has an overpass time of 9:30 am/pm local-time. Since skin temperature, particularly over the land surfaces vary strongly in space and time (Prata et al., 1995), inter-comparison between IASI and MODIS or AIRS, with a time difference of 1 to more than 4 hours can imply a difference of the order of 10 degrees or more in some regions. This makes inter-comparison with other satellite products with different crossing time very difficult to achieve. Moreover, considering IASI's pixel area to be a circle of $\pi \times 12 \times 12$ km$^2$ at nadir and an ellipse





with an area up to π x 20 x 39 km² at its outermost viewing angle of 48° (off-nadir),
several surface types with varying skin temperature and emissivities will co-exist
within one pixel. The resulting skin temperature is therefore an "effective" measure of
the average of the surface-heterogeneity existing in the pixel. This alone complicates
the physical understanding of the $T_{skin}$ values retrieved from space from different
instruments with different pixel shapes (round/ellipse vs square/rectangle, etc.), and
sizes. Moreover, the satellite viewing angle also a role in the $T_{skin}$ at the surface: the
comparison is affected by the different Sun–surface–instrument geometries, as a
result of shadows due to orography or vegetation for example (August et al., 2012).
Finally, the scarcity of in situ $T_{skin}$ ground-observations impedes proper validation,
which in turn is difficult to be properly performed since ground observation is usually
taken at one specific location and time. Given that $T_{skin}$ might strongly change within
short distances (less than a meter, Li et al., 2013), co-locating a satellite
measurement with a ground observation, as we attempted in section 3.3, might
undergo similar large differences as well. Here, a comparison was made at a station
located in a homogenous area to overcome this problem.
Using channel selection and artificial neural network, this work shows a $T_{skin}$ retrieval
method that can serve as a baseline for constructing the first homogeneous dataset
of skin temperature from IASI, and can be extended to other infrared remote
measurements. Future work will look at constructing a $T_{skin}$ time series from IASI
during 2007-present and using Metop A, B, and C for climate trends application.
Regional and seasonal variations can be studied using the atlas for the surface skin
temperature distributions. The daily/monthly/yearly variations will be studied in terms
of the main climate drivers (solar, volcanic eruptions, aerosols and greenhouse
gases) and modes of variability at the inter-annual and decadal timescales.
***Data availability***
The IASI Level 1C data for 2018 are distributed in near real time by Eumetsat
through the EumetCast system distribution. The reprocessed Metop-A L1C data used
in this study for June 2016 will be available in summer 2019 (doi:
10.15770/EUM_SEC_CLM_0014). The EUMETSAT L2 data used in this study can
be retrieved from the Aeris data infrastructure (https://www.aeris-data.fr/). ERA5 data
is provided by ECMWF and can be retrieved at http://www.ecmwf.int or
https://cds.climate.copernicus.eu/. The 12-minute simulation output used in this work
can be obtained by contacting the lead author (sarah.safieddine@latmos.ipsl.fr). The
hourly LST data derived from SEVIRI/Meteosat are freely available from
http://landsaf.ipma.pt within the context of the LSA SAF project funded by
EUMETSAT. The ground observation data can be obtained by contacting F.G.
(frank.goettsche@kit.edu).
***Author contribution***
S.S. wrote the paper with comments from the rest of the co-authors and performed
the neural network calculation and validation. A.P. provided the ERA5/IASI data





matching, M.G. provided data for the ANN training, F.A. and V.P. provided the codes for the channel selection using the ER method, L.C. and S.W. helped in conceptualizing the neural network approach, O.L. provided data for Figure 1, J.N.T, H.H, and G.R. provided the 12-minute ERA5 fields, F.G. and M.M. provided ground measurement data. M. D.-B., D. C. and T. A. helped with the IASI L1C retrieval. C.C. supervised this work and helped with the conceptualization of the study.

***Competing interests***

The authors declare that they have no conflict of interest.

***Acknowledgments***

The authors thank J. Hadji-Lazaro, J.M. Sabater, C. Jimenez, I. Trigo, C.F. Barroso, and P. Kasibhatla for useful discussions. This work was supported by the CNES. It is based on observations with IASI embarked on Metop. The authors acknowledge the Aeris data infrastructure (https://www.aeris-data.fr/) for providing access to the IASI Level 1C data and Level 2 temperature data used in this study. The LST validation site is supported by the Satellite Application Facility (SAF) on Land Surface Analysis (LSA), a European project initiated and financed by EUMETSAT. This project has received funding from the European Research Council (ERC) under the European Union's Horizon 2020 and innovation programme (grant agreement No 742909).

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
