# Peer review of "Skin temperature from the Thermal Infrared Sounder IASI"

_Atmospheric Measurement Techniques, 2019_

## Referee Comment (RC1) · Anonymous Referee #1 · 5 Jul 2019

This paper presents a methodology to retrieve skin temperature from IASI observations using a neural network approach. The channels and retrieval methods seem scientifically correct; however, I'm concerned with the calibration procedure. The authors have chosen skin temperature datasets from EUMETSAT and ERA5, which is acceptable. However, if I understood correctly, the NN is then trained using direct IASI observations. This is not an appropriate procedure since it will result in a NN that is biased towards the datasets used for the retrieval (EUMETSAT and ERA5). This is clear in Fig.5, where the comparison with the ERA5 show the lowest biases. The common procedure is to use a database of atmospheric profiles (from ERA5, for instance) together with a Radiative Transfer Model in order to obtain the best estimates of the relationship between top-of-atmosphere brightness temperatures and skin temperature. This is the

procedure generally used in all currently available operational products. The calibration database is of very high importance in statistical retrieval methods. As such, although the methods are sound, the calibration database is not and I believe it will significantly impact the quality of your retrievals.

There is also no reference to how the authors deal with emissivity. If I understood correctly, you simply disregard it, which means that there will possibly be strong discrepancies between different land covers. Please the discuss the implications of this simplification.

Regarding the inter-comparison and validation exercises, please provide more details on how the spatial matching is performed? Is SEVIRI resampled to the other products resolution or do you use the closest pixel? For the in situ validation a single month does not seem enough to properly validate the products. At least different times of year should be considered. The differences you found for SEVIRI are significantly higher than what was previously reported by Gottshe et al. (2016), how do you justify this? You could have also used SEVIRI to access the spatial variability of the site, e.g. through the std of all SEVIRI pixels within an ISASI observation. Also, in the validation report of EUMETSAT product (EUM/TSS/REP/13/684650), they found that because they were using an area quite far from the station (as you are) sometimes the station area as clouded while the satellite footprint was clear. You might want to use SEVIRI to remove observations when the station is under clouds.

Despite the constrains related the spatial resolution, the authors could also have performed station comparisons with other KIT and e.g. SURFRAD stations as they could provide further information on the quality of the retrievals. Jimenez et al. (2017), for instance, used these to validate retrievals from the AMSR-E, which has approximately the same spatial resolution.

---

## Referee Comment (RC2) · Anonymous Referee #2 · 18 Jul 2019

The manuscript attempts at developing a novel algorithm to derive skin temperature from the IASI sounder via a neural network techniques. Two trainings are chosen to this scope and the results are compared against each other and a third independent in situ source.

My major comment on this manuscript is abouot the conclusion remarks where it is stated that this technique provides a simple method to derive skin temperature from the full IASI constellation. This is true as long as the radiance measurement series is calibrated uniformly and consistently with the training radiance data set. At this stage this uniformly reprocessed radiance data set is missing. Perhaps the authors should aim at developing a set of coefficients for each intermediate time series, especially considering that instrument dis-homogeneities will always be present. More emphasis

should be put to actually explain what is the advantage of this method over the existing EUMETSAT L2 Tskin method.

A concern is the fact that the NN technique seems to strongly depend on the training ensemble. What's the author's take on the impact that this aspect might have on future applications of their data record?

Finally, the author should provide more information about the in situ measurement station. Is this part of an operational network? What type of skin temperature measurement does it exactly perform? Comparing against one single station is reductive in terms of a final assessment of the proposed algorithm. Could more stations be added to the assessment?

On a final note, few additional comments. 1. Few references are missing. The AIRS v6 algorithm employs a NN algorithm to regress skin temperature, along with temperature and water vapor profiles for the AIRS sounder. 2. Besides Ventress and Dudhia, Gambacorta and Barnet 2013, Methodology and information content of the NOAA/NESDIS operational channel selection for infrared hyper spectral sounders, IEEE Geoscience and Remote Sensing Letters, also address the the cross-interference of unwanted species using an initial climatology and updating this with actual retrieval error estimates in a sequential retrieval method. 3. What cloud filtering technique was used to select IASI clear sky radiances in the training?

---

## Author Comment (AC1) · 2 Sep 2019

*We thank Review#1 for their comments. In red italic are our responses to each of the comments*

1- This paper presents a methodology to retrieve skin temperature from IASI observations using a neural network approach. The channels and retrieval methods seem scientifically correct; however, I'm concerned with the calibration procedure. The authors have chosen skin temperature datasets from EUMETSAT and ERA5, which is acceptable. However, if I understood correctly, the NN is then trained using direct IASI observations. This is not an appropriate procedure since it will result in a NN that is biased towards the datasets used for the retrieval (EUMETSAT and ERA5). This is clear in Fig.5, where the comparison with the ERA5 show the lowest biases. The common procedure is to use a database of atmospheric profiles (from ERA5, for instance) together with a Radiative Transfer Model in order to obtain the best estimates of the relationship between top-of-atmosphere brightness temperatures and skin temperature. This is the procedure generally used in all currently available operational products. The calibration database is of very high importance in statistical retrieval methods. As such, although the methods are sound, the calibration database is not and I believe it will significantly impact the quality of your retrievals.

*Yes, the calibration database is very important for a statistical procedure. Using a radiative transfer model to build this database is one of the ways we can construct it. We call this procedure a « physical » database. Another approach is to use an « empirical » database where real satellite observations are put in coincidence with direct observations (radiosondes, buoys, etc…). The authors of this paper have proposed long time ago a calibration dataset based on reanalysis outputs such as in the work done by Aires et al. 2005; Kolassa et al., 2013; and Rodriguez-Fernandez et al., 2015. They have shown that when doing this, it is possible to obtain a satellite retrieval that has no global bias with the reanalysis, but can have strong regional biases with it. The retrieval, even if trained with the reanalysis, does not reproduce the reanalysis, the time and spatial variations are driven by the satellite observations.*

*Such approach is also very interesting if we want to assimilate the retrieved parameter into the reanalysis. This approach has recently been implemented and tested at ECMWF (Rodriguez-Fernandez et al., 2019). The authors have shown that this procedure was an improvement over the classical inversion that used a radiative transfer. This is obviously not always the case but shows the pertinence of the approach. Please note that our procedure is not operationally used at ECMWF.*

*We addressed the Reviewer's comment, by adding the following sentence when discussing ANN in section 2.3 as follows:*

*"The feasibility of using ANN to $T_{skin}$ retrieval has been shown for instance by Aires et al. (2002) for IASI, and has also been performed to tackle various problems in atmospheric remote sensing (Blackwell and Chen, 2009; Hadji-Lazaro et al., 1999; Whitburn et al., 2016; Van Damme et al., 2017). **The retrieval, even if trained with the reanalysis, does not reproduce the reanalysis; the time and spatial variations are driven by the satellite observations (Aires et al. 2005; Kolassa et al., 2013; Rodriguez-Fernandez et al., 2015**)."*

2- There is also no reference to how the authors deal with emissivity. If I understood correctly, you simply disregard it, which means that there will possibly be strong discrepancies between different land covers. Please the discuss the implications of this simplification.

*The authors of this paper are well aware of the complexity to deal with emissivity. They have been working on surface emissivities (MW, IR over land and ocean) for the last two decades (Aires et al., 2011; Paul et al., 2012). The latter study (Paul et al., 2012) shows a complex physical scheme to simultaneously retrieve Ts and surface emissivities. In this study, as the goal is to look at temperature changes trends for specific locations, we intend to present a simpler approach where information on emissivity is assumed to be included in the radiance spectra since it is quite exhaustive (see the conclusions of the study). The results and comparison on a global scale are very encouraging, although we agree that it probably affects validation locally. We will look into including information on emissivities in the future.*

3- Regarding the inter-comparison and validation exercises, please provide more details on how the spatial matching is performed? Is SEVIRI resampled to the other products resolution or do you use the closest pixel?

*The nearest pixel is chosen. It is mentioned in section 2.4.3: "IASI and SEVIRI data are spatially co-located when observations from each instrument are less than 5 minutes apart, and within 0.25 degrees in longitude and latitude."*

4- For the in situ validation a single month does not seem enough to properly validate the products. At least different times of year should be considered. The differences you found for SEVIRI are significantly higher than what was previously reported by Gottshe et al. (2016), how do you justify this? You could have also used SEVIRI to access the spatial variability of the site, e.g. through the std of all SEVIRI pixels within an ISASI observation.

*Generally speaking, it is hard to validate satellite measurements with ground LST given that the footprint of the satellite instrument will have various land surface types and the LST will therefore be an effective measure of this surface inhomogeneity. Gobabab is therefore uniquely suitable for validating IASI LST because of the large homogenous areas around it, as figure 7 (hereafter) panel (b) shows. To extend our analysis and to address the Reviewer concerns (also a concern for the second Reviewer), we performed a validation over the whole year (instead of just one month). The results and discussion show similar results to the one-month validation, as the figure hereafter shows:*

[Figure]

*New* **Figure 7. Comparison of IASI $T_{ANN}$ with ground observations at Gobabeb: (a) Diurnal and seasonal variation of $T_{skin}$; (b) station and validation site location with a one-month example of IASI-coincident observations; (c) $T_{ANN}$ versus in-situ $T_{skin}$ during the day; and (d) during the night for all coincident observations in 2016.**

*The discussion of the figures has been updated in different locations in the main text, and we point out that the conclusions are not very different from the original ones.*

5- Also, in the validation report of EUMETSAT product (EUM/TSS/REP/13/684650), they found that because they were using an area quite far from the station (as you are) sometimes the station area as clouded while the satellite footprint was clear. You might want to use SEVIRI to remove observations when the station is under clouds.

*It won't be possible to use clear-sky SEVIRI measurements to choose clear-sky measurements at the site, because the validation is done at the crossing-time of IASI, which might not correspond to the crossing time of SEVIRI at the site.*

*We think that the Reviewer brings a good point, so we discuss this in the text at the end of section 3:* **"Differences between the different products and ground measurements can be due to cases where the sky at the in-situ measurement site is at least partly**

**cloudy/clear, while being clear/partly cloudy at the validation site (EUMETSAT, 2013)."**

6- Despite the constrains related the spatial resolution, the authors could also have performed station comparisons with other KIT and e.g. SURFRAD stations as they could provide further information on the quality of the retrievals. Jimenez et al. (2017), for instance, used these to validate retrievals from the AMSR-E, which has approximately the same spatial resolution.

*We extend the validation around Gobabeb to a whole year, as discussed in point #4 in this review.  However, to answer the Reviewer specific comment, we'd like to point out that AMSR-E Tskin retrieval is placed in the 14×8 km$^2$ (=112 km$^2$) swath grid of the 36.5 GHz channel. IASI's pixel area is at best a circle of π x 6 x 6 km$^2$ at nadir (=113 km$^2$) and an ellipse with an area up to ~ π x 10 x 20 km$^2$ at its outermost viewing angle of 48° (off-nadir).*

*The SUFRAD stations shown hereafter are all around inhomogeneous land surface types. A 12 km ruler is placed over each of the SUFRAD locations to show the minimal IASI pixel (which would be at best 4 pixels out of the 120 pixels in one swath, without cloud filtering which usually takes out 2/3 of measurements). Clearly, from the pictures, many land types exist around the validation sites, which complicates the validation.*

*All SURFRAD stations deliver long-term measurements of the surface radiation budget. This is done by measuring downwelling and upwelling broadband solar and thermal infrared (TIR) irradiance. Skin temperature has to be derived from incoming and outgoing IR radiance measurements, and by estimating emissivities. We therefore believe that this would introduce many sources of error into the comparison with IASI since the emissivity is a function of land type that changes over the IASI's pixel. We therefore only use in our discussion/validation the Gobabeb station. Upon discussing with the KIT's stations PI, we realized again that the horizontal resolution is an issue. Gobabab is the only of KIT's site that is suitable for validating IASI LST: the homogenous areas around the other sites are just too small.*

[Figure]

*The location of the different SUFRAD stations. The white arrows are the best/Nadir IASI observations. It is clear from these pictures that validation around these sites is challenging. Source: Google Maps.*

**References added**

*Aires, F., Prigent, C., and Rossow, W.B.: Soil moisture at a global scale. II – Global statistical relationships. JGR, 110, D11, D11103, 10.1029/2004JD005094, 2005.*

*Aires, F., Prigent, C., Bernardo, F., Jimenez, C., Sounders, R., and Brunel, P.: A Tool to Estimate Land Surface Emissivities in the Microwaves (TELSEM) for use in numerical weather prediction schemes. QJRMS, 137: 690-699, 10.1002/qj.803, 2011.*

*EUMETSAT: IASI L2 Metop-B Validation Report, v3A, 19 July 2013, EUM/TSS/REP/13/684650, https://www.eumetsat.int/website/home/Data/TechnicalDocuments/index.html, 2013.*

*Kolassa, J., Aires, F., Polcher J., Prigent, C., Jimenez, C., and Pereira, J.M.: Soil moisture retrieval from multi-instrument observations: Part I – Information content analysis and retrieval methodology, JGR, 118, 10, 4847-4859, 10.1029/2012JD018150, 2013.*

*Paul, M., Aires, F., and Prigent, C.:An innovative physical scheme to retrieve simultaneously surface temperature and emissivities based on a high-resolution infrared emissivity interpolator, JGR, 117, D11302, 10.1029/2011JD017296, 2012.*

*Rodriguez-Fernandez, N.J., Aires, F., Richaume, P., Cabot, F., Jimenez, C., Kerr, J., Kolassa, J., Mahmoodi, A., Prigent, C., and Drush, M.: Soil moisture retrieval from SMOS observations using neural networks, IEEE TGRS, 53, 11, 10.1109/TGRS.2015.2430845, 2015.*

*Rodríguez-Fernández, N., de Rosnay, P., Albergel, C., Richaume, P., Aires, F., Prigent, C., Kerr, Y.: SMOS Neural Network Soil Moisture Data Assimilation in a Land Surface Model and Atmospheric Impact,* Remote Sens. 11*(11),* 1334; *https://doi.org/10.3390/rs11111334, 2019.*

---

## Author Comment (AC2) · 2 Sep 2019

*We thank Review#2 for their comments that helped improving the study. In red italic are our responses to each of the comments*

**The manuscript attempts at developing a novel algorithm to derive skin temperature from the IASI sounder via a neural network techniques. Two trainings are chosen to this scope and the results are compared against each other and a third independent in situ source.**

1- **My major comment on this manuscript is abouot the conclusion remarks where it is stated that this technique provides a simple method to derive skin temperature from the full IASI constellation. This is true as long as the radiance measurement series is calibrated uniformly and consistently with the training radiance data set. At this stage this uniformly reprocessed radiance data set is missing. Perhaps the authors should aim at developing a set of coefficients for each intermediate time series, especially considering that instrument dis-homogeneities will always be present. More emphasis should be put to actually explain what is the advantage of this method over the existing EUMETSAT L2 Tskin method.**

*The radiance dataset used in this study is uniformly reprocessed. This is mentioned in the introduction here:*

*"The Metop-A L1C record has been reprocessed back in time at EUMETSAT for the period 2007-2017, and is used in this work, and will be publically available in summer 2019. L1C data after 2017 are not reprocessed because they are assumed to be up to date and consistent with reprocessed data. The Level 2 series has not yet been reprocessed back in time, which complicates the construction of a homogeneous $T_{skin}$ data record from IASI."*

*As such we were the first ones to use this homogeneous radiances dataset to produce a consistent Tskin data record. It is worth mentioning that the Eumetsat L2 products (clouds, Tskin and T profiles, trace gas contents) are evolving with time and algorithm improvements. No backward processing was done so far, so there is no Tskin record available for climate studies. That's the advantage of this work: providing the only homogeneous Tskin data record from IASI.*

*We address the Reviewer comment, by reminding the reader in the conclusion that the L1C data record is homogenous as follows:*

*"[… ] Consequently, no homogenous consistent IASI $T_{skin}$ record exists to date.*
***However, in this study we take advantage of the fact that the Metop-A L1C radiances, recently reprocessed at EUMETSAT and used in this work, are homogeneous."***

2- **A concern is the fact that the NN technique seems to strongly depend on the training ensemble. What's the author's take on the impact that this aspect might have on future applications of their data record?**

*Indeed, the NN will depend on the training ensemble. However, the aim of this work is to have a Tskin a homogeneous product from IASI to analyze the regional and global temperature distributions. Our algorithm is not as sophisticated as the EUMETSAT L2 algorithm or the ECMWF reanalysis, and depends on them. However, it also depends on the IASI radiances, and those are changing independently.*

*are put in coincidence with direct observations (radiosondes, buoys, etc…). The authors of this paper have proposed long time ago a calibration dataset based on reanalysis outputs such as in the work done by Aires et al. 2005; Kolassa et al., 2013; and Rodriguez-Fernandez et al., 2015. They have shown that when doing this, it is possible to obtain a satellite retrieval that has no global bias with the reanalysis, but can have strong regional biases with it. The retrieval, even if trained with the reanalysis, does not reproduce the reanalysis, the time and spatial variations are driven by the satellite observations.*

*We addressed the Reviewer's comment, by adding the following sentence when discussing ANN in section 2.3 as follows:*

"The feasibility of using ANN to T$_{skin}$ retrieval has been shown for instance by Aires et al. (2002) for IASI, and has also been performed to tackle various problems in atmospheric remote sensing (Blackwell and Chen, 2009; Hadji-Lazaro et al., 1999; Whitburn et al., 2016; Van Damme et al., 2017). **The retrieval, even if trained with the reanalysis, does not reproduce the reanalysis; the time and spatial variations are driven by the satellite observations (Aires et al. 2005; Kolassa et al., 2013; Rodriguez-Fernandez et al., 2015).**"

**3-  Finally, the author should provide more information about the in situ measurement station. Is this part of an operational network? What type of skin temperature measurement does it exactly perform?**

*We provide more information on the in situ measurement station, by expanding section 2.4.4 as follows:*

"The ground observations are from Gobabeb wind tower, Namibia (23.551° S 15.051° E, location shown in Figure 7, Göttsche et al., 2016). Gobabeb station is located on the large and homogenous Namib gravel plains (Göttsche and Hulley, 2012). **It is part of the Karlsruhe Institute of Technology (KIT) stations, designed for continuous validation of LST products over several years. The core instruments of KIT's validation stations are Heitronics KT15.85 IIP infrared radiometers that measure radiances between 9.6 and 11.5 µm. The temperature resolution is 0.03 K with an uncertainty of ±0.3 K over the relevant range, and high stability with a drift of less than 0.01 % per month (Goettsche et al., 2013). Based on in-situ measurements, the surface emissivity of the gravel plains is estimated as 0.944 +-0.015 for MSG/SEVIRI 10.8 µm channel (Göttsche and Hulley, 2012). During an international inter-comparison campaign in-situ emissivity spectra were obtained at 49 sample**

*locations distributed across the gravel plains: the results confirm the previously obtained results (Göttsche et al., 2018).*
*The IR radiance measurements from KIT stations have been successfully used to validate several satellite LST products derived from MODIS (Freitas et al., 2010; Guillevic et al., 2013; Ermida et al., 2014), SEVIRI (Freitas et al., 2010; Goettsche et al., 2013; Ermida et al., 2014) and a range of sensors (Martin et al., 2019). The monitoring capability of KIT's validation stations was demonstrated by Göttsche et al. (2016) for LST derived from MSG/SEVIRI[..]"*

4- **Comparing against one single station is reductive in terms of a final assessment of the proposed algorithm. Could more stations be added to the assessment?**

*Generally speaking, it is hard to validate satellite measurements with ground LST given that the footprint of the satellite instrument will have various land surface types and the LST will therefore be an effective measure of this surface inhomogeneity. Gobabab is the only of KIT's site that is suitable for validating IASI LST: the homogenous areas around the other sites are just too small. To extend our analysis and to address Reviewer#1 concerns, we perform a validation over the whole year (instead of just one month). The results and discussion show similar results to the one-month validation, as the figure hereafter shows:*

[Figure]

*New **Figure 7. Comparison of IASI TANN with ground observations at Gobabeb: (a) Diurnal and seasonal variation of Tskin; (b) station and validation site location with a one-month example of IASI-coincident observations; (c) TANN versus in-situ Tskin during the day; and (d) during the night for all coincident observations in 2016.***

*The discussion of this figure is updated in the manuscript when necessary. Since it doesn't change much from the conclusions of the comparison with the one-month data, we don't include it here and we ask the Reviewer to refer to the corrected version of the paper if needed.*

**5-  On a final note, few additional comments. 1. Few references are missing. The AIRS v6 algorithm employs a NN algorithm to regress skin temperature, along with temperature and water vapor profiles for the AIRS sounder.**

*Two references related to AIRS NN v6 were added when discussing ANN and Tskin retrievals in the introduction as follows:*

"The feasibility of using ANN to T$_{skin}$ retrieval has been shown for instance by Aires et al. (2002) for IASI, and has also been performed to tackle various problems in atmospheric remote sensing (Blackwell and Chen, 2009; Hadji-Lazaro et al., 1999; Whitburn et al., 2016; Van Damme et al., 2017). [..] ***For AIRS and AMSU, projected principal***

*components for coefficient compression and a neural network trained using global training set derived from European Center for Medium-Range Weather Forecasting (ECMWF) fields are used in the version 6 retrievals of atmospheric temperatures and water vapor (Milstein and Blackwell, 2016; Tao et a., 2013)."*

6- **Besides Ventress and Dudhia, Gambacorta and Barnet 2013, Methodology and information content of the NOAA/NESDIS operational channel selection for infrared hyper spectral sounders, IEEE Geoscience and Remote Sensing Letters, also address the the cross-interference of unwanted species using an initial climatology and updating this with actual retrieval error estimates in a sequential retrieval method.**

*The reference was included along with Ventress and Dudhia.*

7- **What cloud filtering technique was used to select IASI clear sky radiances in the training?**

*Cloud filtering from AVHRR on Metop was used. This was added when discussing cloud free radiances in section 2.3 as follows:*

"The training dataset is constructed out of clear-sky scenes (cloud cover <10%) *selected* ***using AVHRR measurements, collocated with those of IASI on Metop (Maddy et al., 2011).*** Level 1C (L1C) clear-sky IASI radiances are used over the 100 channels selected in section 2.2."

***References added***

*Aires, F., Prigent, C., and Rossow, W.B.: Soil moisture at a global scale. II – Global statistical relationships. JGR, 110, D11, D11103, 10.1029/2004JD005094, 2005.*

*Ermida, S.L., Trigo, I.F., DaCamara, C.C., Göttsche, F.-M., Olesen, F.-S. and Hulley, G.: Validation of remotely sensed surface temperature over an oak woodland landscape— The problem of viewing and illumination geometries. Remote Sensing of Environment, 148, 16-27, 2014.*

*Gambacorta A. and Barnet C.D., Methodology and Information Content of the NOAA NESDIS Operational Channel Selection for the Cross-Track Infrared Sounder (CrIS), IEEE T. Geosci. Remote, 51, 3207–3216, 2013.*

*Guillevic, P. C., A. Bork-Unkelbach, F. M. Gottsche, G. Hulley, J.-P. Gastellu-Etchegorry, F. S. Olesen, and J. L. Privette: Directional Viewing Effects on Satellite Land Surface Temperature Products Over Sparse Vegetation Canopies—A Multisensor Analysis. IEEE Geosci. Remote Sens. Lett., 10, 1464–1468, doi:10.1109/LGRS.2013.2260319, 2013.*

*Göttsche, F.-M., Olesen, F., Poutier, L., Langlois, S., Wimmer, W., Santos, V.G., Coll, C., Niclos, R., Arbelo, M., and Monchau, J.-P.: Report from the field inter-*

comparison experiment (FICE) for land surface temperature, http://www.frm4sts.org/project-documents/, Accessed 01/09/2019, 2018.

Kolassa, J., Aires, F., Polcher J., Prigent, C., Jimenez, C., and Pereira, J.M.: Soil moisture retrieval from multi-instrument observations: Part I – Information content analysis and retrieval methodology, JGR, 118, 10, 4847-4859, 10.1029/2012JD018150, 2013.

Maddy, E. S., King, T. S., Sun, H., Wolf, W. W., Barnet, C. D., Heidinger, A., Cheng, Z., Goldberg, M. D., Gambacorta, A., Zhang, C., and Zhang, K.: Using MetOp-AAVHRR Clear-Sky Measurements to Cloud-ClearMetOp-AIASI Column Radiances, J. Atmos. Ocean. Tech., 28, 1104–1116, doi:10.1175/jtech-d-10- 05045.1, 2011.

Milstein, A. B., and Blackwell, W. J.: Neural network temperature and moisture retrieval algorithm validation for AIRS/AMSU and CrIS/ATMS,J. Geophys. Res. Atmos.,121, 1414–1430,doi:10.1002/2015JD024008, 2016.

Rodriguez-Fernandez, N.J., Aires, F., Richaume, P., Cabot, F., Jimenez, C., Kerr, J., Kolassa, J., Mahmoodi, A., Prigent, C., and Drush, M.: Soil moisture retrieval from SMOS observations using neural networks, IEEE TGRS, 53, 11, 10.1109/TGRS.2015.2430845, 2015.

Tao, Z., Blackwell, W. J., and Staelin, D. H.: Error Variance Estimation for Individual Geophysical Parameter Retrievals, IEEE Transactions on Geoscience and Remote Sensing, vol. 51, no. 3, pp. 1718-1727, doi: 10.1109/TGRS.2012.2207728, 2013.